# Phytocompounds from Amazonian Plant Species against Acute Kidney Injury: Potential Nephroprotective Effects

**DOI:** 10.3390/molecules28176411

**Published:** 2023-09-02

**Authors:** Alberto Souza Paes, Rosemary de Carvalho Rocha Koga, Priscila Faimann Sales, Hellen Karine Santos Almeida, Thiago Afonso Carvalho Celestino Teixeira, José Carlos Tavares Carvalho

**Affiliations:** 1Pharmaceutical Innovation Program, Department of Biological and Health Sciences, Federal University of Amapá, Rodovia Juscelino Kubitschek, km 02, Macapá CEP 68903-419, Amapá, Brazil; dralbertopaes343ap@hotmail.com (A.S.P.); rosemarykoga@unifap.br (R.d.C.R.K.); pfaimann@gmail.com (P.F.S.); thafonsoteixeira@gmail.com (T.A.C.C.T.); 2Research Laboratory of Drugs, Department of Biological and Health Sciences, Federal University of Amapá, Rodovia Juscelino Kubitschek, km 02, Macapá CEP 68903-419, Amapá, Brazil; hellenkarine1612@gmail.com; 3University Hospital, Federal University of Amapá, Rodovia Josmar Chaves Pinto, km 02, Macapá CEP 68903-419, Amapá, Brazil

**Keywords:** Amazonian traditional medicine, phytotherapy, hypoxia, oxidative stress, nephroprotection, antioxidant, anti-inflammatory, diuretic

## Abstract

There are several Amazonian plant species with potential pharmacological validation for the treatment of acute kidney injury, a condition in which the kidneys are unable to adequately filter the blood, resulting in the accumulation of toxins and waste in the body. Scientific production on plant compounds capable of preventing or attenuating acute kidney injury—caused by several factors, including ischemia, toxins, and inflammation—has shown promising results in animal models of acute kidney injury and some preliminary studies in humans. Despite the popular use of Amazonian plant species for kidney disorders, further pharmacological studies are needed to identify active compounds and subsequently conduct more complex preclinical trials. This article is a brief review of phytocompounds with potential nephroprotective effects against acute kidney injury (AKI). The classes of Amazonian plant compounds with significant biological activity most evident in the consulted literature were alkaloids, flavonoids, tannins, steroids, and terpenoids. An expressive phytochemical and pharmacological relevance of the studied species was identified, although with insufficiently explored potential, mainly in the face of AKI, a clinical condition with high morbidity and mortality.

## 1. Introduction

Natural products have been a source of important biologically active substances, and this is due to the diversity of chemical compounds that can be found in plants, fungi, and bacteria, among other organisms [1]. These compounds have a wide variety of structures and biological activity, which makes them potential candidates for the development of new drugs [2]. Products derived from plant species, in particular, those from the Amazonian biodiversity, are a rich source of compounds with promising pharmacological properties, including nephroprotective, anti-inflammatory, and antioxidant activity [3].

Some of the classes of Amazonian plant compounds have been investigated for their potential activity in kidney protection, among them, plant species belonging to the alkaloid classes, for their nephroprotective effects in experimental models of acute kidney injury (AKI); flavonoids, due to antioxidant and anti-inflammatory properties with renal protective activity; tannins, due to their antioxidant and anti-inflammatory potential, with probable prevention or reduction of kidney damage induced by toxins and other substances; steroids, with anti-inflammatory and antioxidant activity, which may help to prevent acute kidney injury induced by oxidative stress; and terpenoids, for their potent anti-inflammatory, antioxidant, and immunomodulatory activity [4,5,6].

Therefore, knowing this Amazonian biodiversity allows us to exploit it in the best way and protect it. This is mainly due to great expectations regarding the environmentally correct exploitation of natural resources for production and processing and a fair return for the traditional population [7].

These natural compounds can be isolated and used as models for the development of new drugs, or they can be chemically modified to improve their pharmacokinetic and pharmacodynamic properties, resulting in new compounds with potential therapeutic activity against AKI [8,9,10]. AKI is a medical condition that can lead to acute renal failure, with a high risk of morbidity and mortality [11]. AKI is characterized by an abrupt reduction in renal function, with accumulation of toxic metabolites and electrolytes in the body, triggering serious complications such as hemodynamic disorders, pulmonary edema, metabolic disorders, and even death [12].

AKI can affect up to 7% of hospitalized patients and up to 50% of critically ill patients, being one of the main causes of mortality in this group of patients. The dysfunction comes from a variety of factors, including the use of nephrotoxic drugs, renal ischemia, heart failure, hypovolemia, and sepsis, among others [13]. Advanced age, the presence of comorbidities such as diabetes and hypertension, and the use of invasive procedures such as cardiac surgery are also associated with a higher risk of developing AKI [12].

This clinical condition generates several social problems, especially in countries with precarious health systems, resulting in (a) increased health costs, due to the need for intensive and prolonged treatment, including intensive care, and, in some cases, dialysis; (b) reduced quality of life, especially in the most serious cases, where recovery can be slow and complicated, resulting in loss of productivity, inability to work, and the need for special care; (c) socioeconomic inequalities, disproportionately affecting more vulnerable populations, including elderly patients, people with low socioeconomic status, or those with limited access to adequate health care; (d) overload of healthcare systems, especially in countries with limited resources, due to the requirement for intensive and long-term care [14].

AKI involves a series of complex mechanisms, including hemodynamic disturbances, in which there is a reduction in renal perfusion, with consequent hypoxia and ischemia, resulting in cell damage and inflammation [12]. AKI is often accompanied by kidney inflammation, which can contribute to cell damage and tubular dysfunction. This reduction in kidney function with damage to the epithelial cells of the renal tubules, releases toxic metabolites and electrolytes into the body, resulting in serious complications. In addition to tubular dysfunction, it can reduce the ability to reabsorb water and electrolytes as well as the ability to excrete toxic metabolites [13].

Based on its pathophysiology, it is classified into three types: pre-renal, due to a reduction in renal perfusion due to hypovolemia, heart failure, or hypotension, resulting in hypoxia and cell damage; intrinsic, due to direct damage to the kidneys, including nephrotoxicity from medications, exposure to toxic chemicals, infections, or autoimmune diseases; postrenal, caused by kidney stones, tumors, or other obstructions to the flow of urine, resulting in accumulation of urine in the kidneys and subsequent damage [15].

Therefore, AKI can be potentially fatal if not treated properly, and conventional treatments often have significant side effects [12]. In this context, the Amazonian population uses several plant species with bioactive compounds to treat diseases of the renal and urinary system [16]. Traditional Amazonian medicine has contributed to the discovery of new bioactive products [1]. Therefore, developing drugs from plant species from the Amazon against AKI can help combat resistance to existing drugs. This can also minimize significant side effects, including damage to the liver and other organs; improve effectiveness; and provide new treatment options for AKI.

## 2. Method

This study constitutes, methodologically, an analytical bibliographic review related to the mapping of secondary metabolites found in Amazonian plant species with potential to treat AKI, belonging to the classes of alkaloid compounds, flavonoids, tannins, terpenoids, and steroids. Among the species highlighted are *Banisteriopsis caapi* (Spruce ex Griseb.) Morton, *Peganum harmala* L., *Passiflora edulis* Sims, *Annona muricata* L., *Uncaria tomentosa* (Willd.) DC., *Hymenaea courbaril* L., *Echinodorus macrophyllus* (Kunth) Micheli, *Acmella oleracea* (L.) R. K. Jansen, and *Rosmarinus officinalis* L., in addition to studies on potential nephroprotective effects. Data collection was carried out from September 2022 to February 2023, using the following databases: CAPES journals, PubMed, Science Direct from Elsevier, Wiley Online Library, Springer-Nature, Taylor and Francis, BMC, Hindawi, Scielo, ACS—American Chemical Society, and Google Scholar, as well as databases of scientific articles and patents “The LENS” and “ORBIT Intelligence”.

The inclusion criteria for this work included original articles exclusive to the genus and species studied, with full text available in Portuguese, English, and other languages. Exclusion criteria included abstracts, online sites without scientific sources, incomplete texts, and unrelated and repeated articles.

As for the search strategy, the descriptive words used in this work were as follows: species *Banisteriopsis caapi* (Spruce ex Griseb.) Morton, *Peganum harmala* L., *Passiflora edulis* Sims, *Annona muricata* L., *Uncaria tomentosa* (Willd.) DC., *Hymenaea courbaril* L., *Echinodorus macrophyllus* (Kunth) Micheli, *Acmella oleracea* (L.) R. K. Jansen, and *Rosmarinus officinalis* L., correlated with secondary metabolites and their nephroprotective potential. The articles were selected by reading the titles and abstracts of the publications, associated with the Boolean descriptor “AND”, in order to refine the samples.

The review is based primarily on articles published after 2010. However, some older articles were also mentioned to provide relevant background or when providing well-documented information. The study shows the expressive phytochemical and pharmacological relevance of the studied species, although many of them with insufficiently explored potential, mainly in the face of AKI.

## 3. Secondary Metabolites and Nephroprotective Potential in Amazonian Plant Species

### 3.1. Classes of Compounds Present in Amazonian Plant Species

#### 3.1.1. Alkaloids

Alkaloids are a class of nitrogenous organic compounds that occur naturally in plant species [17]. Such constituents are a class of chemical compounds with alkaline properties that contain at least one nitrogen atom in their structure, being produced by plants as a form of defense against herbivores and pathogens [18].

Alkaloids are characterized by having a heterocyclic ring structure with at least one nitrogen atom, unlike aliphatic nitrogen compounds, which are non-cyclic [19]. These substances can occur as homoligomeric or heteroligomeric monomers, dimers, trimers, or tetramers. There are two main groups of alkaloids: those with a heterocyclic or non-heterocyclic chemical structure, and those of biological or natural origin, which come from specific sources [20].

Alkaloids can be divided into different classes or groups based on their chemical structures and properties: indole (serotonin, melatonin, and tryptamine), isoquinoline (morphine, codeine, and papaverine), terpenic (atropine, scopolamine, and ephedrine), and pyrrolizidine (senecionin and retronecin) [21].

Many alkaloids have pharmacological properties such as analgesics [22], hallucinogens [23], anesthetics [24], antidiabetic [25], and anticancer [18]. Some studies suggest that plants that produce harmine, harmaline, and tetrahydroharmine alkaloids may have nephroprotective biological activity; that is, they may protect the kidneys against damage [26].

However, it is worth mentioning the toxicity of certain metabolites belonging to the class of alkaloids, for example, saponins. According to Fang et al. [27], in an acute toxicity test, with crude extracts of *Albizia coriaria* (used by the traditional population of Uganda), with a significant percentage of saponins in phytochemical screening, extracts of the species possibly produced acute renal toxicity, based on histopathological identification of tissue with severe acute multifocal nephritis, characterized by infiltration of inflammatory cells at various sites in the renal interstitium. In addition, the animal models used clinically had excessive urination, with a probable link to toxicity.

#### 3.1.2. Flavonoids

Flavonoids are a class of organic compounds widely distributed in nature, characterized by their flavone chemical structure, which includes two aromatic rings joined by a three-carbon bridge. They have been the subject of studies due to the wide range of biological properties derived from their bioactive compounds [28]. Based on their chemical structures and biological properties, flavonoids are divided into several subclasses, such as flavones (a hydroxyl group at the 4 position of the B ring: luteolin and apigenin), flavonols (a hydroxyl group at the 3 position of the C ring: quercetin and kaempferol), flavanones (without a hydroxyl group at position 3 of the C ring: hesperidin and naringin), flavanols (a hydroxyl group at position 3 and a hydroxyl group at position 4 of the C ring: catechin and epicatechin), anthocyanins (water-soluble pigments responsible for the red, purple, and blue colors of many fruits and vegetables), and isoflavones (found primarily in legumes) [29].

Several preclinical and clinical studies have documented the pharmacological activity of flavonoids, mainly their antioxidant properties [30,31], antidiabetics [32], antiobesity [33], antihyperlipidemic [34], anti-inflammatory [35], anti-osteoporotic effects [36], antiallergic, and antithrombotic [37], in addition to being hepatoprotective [38], neuroprotective [39], nephroprotectors [40,41,42], chemopreventives, and anticancers [43], as well as having antibacterial, antifungal, and antiviral activity [44]. Flavonoids can inhibit in vitro proliferation of several cancer cell lines and reduce tumor growth in animal models [43]. They are recognized as antioxidants and have properties that eliminate free radicals. Thus, they act as divalent cation chelators and have free radical scavenging properties, inhibiting lipid peroxidation, capillary permeability, and platelet aggregation and fragility [45].

These compounds are able to increase the activity of endogenous free radical metabolizing enzymes, including catalase (CAT), superoxide dismutase (SOD), glutathione peroxidase, and glutathione (GSH), which are crucial for the elimination of ROS and consequently the increase of antioxidant activity. Several studies have shown that pretreatment with flavonoids can play a key role in ischemia and reperfusion injury [46].

In a study carried out in a renal model of AKI in Wistar rats, pretreatment with rutin significantly reduced renal failure, in addition to inhibiting the production of malondialdehyde (MDA) and restoring depleted levels of GSH and superoxide dis-mutase activity [47]. The compound apigenin increased SOD and glutathione peroxidase activity, as well as being able to reduce MDA in a rat model of AKI through activation of the JAK2/STAT3 signaling pathway [48].

Flavonoids modulate oxidative stress through the pathway of nuclear factor 2 related to erythroid 2 (Nrf2), a transcription factor that regulates the expression of several cytoprotective and antioxidant genes [49]; inhibit the production of peroxynitrite, suppressing the iNOS activity and NO production; and are essential mediators of the pathological and physiological processes of AKI [50].

In addition to promoting an inflammatory response in the biological processes of AKI, they inhibit the tumor necrosis factor-α (TNF-α), which initiates the inflammatory cascade and the positive regulation of chemokines and cytokines such as IL-6 and IL-1β, which can damage renal cells directly [51]. In addition, in toll-like receptor 4 (TLR4) signaling, they are an important modulator of chemokines and pro-inflammatory cytokines, migration and infiltration of leukocytes, and apoptosis of renal tubular epithelial cells [52].

In addition, flavonoids regulate biological systems through the inhibition of several enzymes, including hydrolase, lipase, α-glucosidase, aldose reductase, cyclooxygenase, xanthine oxidase, hyaluronidase, alkaline phosphatase, arylsulfatase, lipoxygenase, Ca^+2^-ATPase, cAMP phosphodiesterase, and various kinases [53].

#### 3.1.3. Tannins

Tannins are the most abundant secondary metabolites produced by plants [54]. Tannins are widespread in the plant kingdom and occur in different concentrations in all parts of plant material, be it bark, fruit, wood, or roots [55]. Vegetable tannins are generally classified into two groups: pyrogallol tannins or hydrolysable tannins and catechol tannins or condensable tannins. Those of the hydrolysable type, in turn, are subdivided into two groups: gallotannins, which produce gallic acid and glucose, and ellagitannins, which provide ellagic acid and glucose. Condensable tannins are not prone to hydrolysis but are amenable to oxidation and polymerization to form insoluble products known as red tannins/phlobaphenes [56].

Tannins have been the subject of several studies due to their potential pharmacological effects. They are known to have antioxidant activity, protecting the body’s cells against oxidative damage caused by free radicals [57]. They promote anti-inflammatory action [58] and are also able to kill or inhibit the growth of bacteria, fungi, and viruses, thus exhibiting antimicrobial activity [59]. Tannins have been studied for their potential antitumor effect by inhibiting the growth and proliferation of tumor cells [60]. They also have hypoglycemic effects in vivo [61] and are excellent promoters of hepatoprotection, against damage caused by toxins and other harmful substances [62]. Among the cardiovascular effects, some studies suggest that tannins can help reduce the risk of cardiovascular diseases, including atherosclerosis and hypertension [63].

In terms of nephroprotection, the tannins obtained from the methanolic extract of the *Jatropha tanjorensis* leaf improved the serum levels of the renal metabolites urea, uric acid, and creatinine in animals exposed to renal damage by sodium benzoate; treatment with the extract reversed significantly changes these important markers of kidney damage in a dose-dependent manner [64].

In Brazil, tea from the plant *Phyllanthus niruri* known as ‘stone breaker’ is commonly used in cases of kidney stones. The species does not present acute or chronic toxicity; it has a uricosuric effect and increases glomerular filtration, which suggests its potential use not only as a lytic and/or preventive effect on the formation of urinary calculi but also as a possible use in hyperuricemic patients (by the uricosuric effect) and patients with renal failure [65]. According to the Brazilian Pharmacopoeia [66], the plant drug is derived from the dried aerial parts of *Phyllanthus niruri* L. [syn. *Phyllanthus niruri* ssp. niruri L. and *Phyllanthus niruri* ssp. lathyroides (Kunth) G.L.Webster] containing at least 6.5% of total tannins and 0.15% of gallic acid (C_7_H_6_O_5_, 170.12) for herbal marketing.

#### 3.1.4. Steroids

Steroids are organic compounds that occur naturally in plants; they have a characteristic sterol ring molecular structure [67]. Each type of steroid has a specific function, playing important roles in the physiology and biochemistry of those plant species in which they are found. For example, phytosterols play an important role in regulating cell membrane permeability while brassinosteroids act as growth and development hormones [68]. Although natural products are often associated with deleterious health effects, plant steroids have many medicinal applications, and research continues to explore these secondary metabolites as potential leaders in drug design and discovery [69].

Several natural steroids have been extensively studied for pharmacological efficacy as antihormones [70], contraceptive drugs [71], anticancer agents [72], cardiovascular agents [73], osteoporosis medications [74], antibiotics, anesthetics, anti-inflammatories and antiasthmatics [75].

Important roles for steroids have been evaluated, in diuresis and renal protection, from secondary metabolites of plant species traditionally used by the Amazonian population [76]. Studies in animal models of nephrotoxicity have shown nephroprotective activity [77], reduction of lipid peroxidation and renal fibrosis, and improvement of renal function in models of acute kidney injury [78].

#### 3.1.5. Terpenoids

Terpenoids constitute the largest class of secondary metabolites and generally do not contain nitrogen or sulfur in their structures. As a consequence, many terpenoids have pronounced pharmacological activity and are therefore interesting for medicine and biotechnology [79].

Terpenoids are classified by the number of five-carbon (isoprene) units they contain. Thus, the smallest terpenes contain a single isoprene, called hemiterpene, monoterpenoids (with two isoprene units), sesquiterpenoids (three units), diterpenoids (four units), triterpenoids (six units), tetraterpenoids (eight units), and polyterpenoids (more than eight units). Each group of terpenoids has distinct physical and chemical properties, which influence their biological and pharmacological activity [80].

Several in vitro, preclinical, and clinical studies have confirmed that this class of compounds exhibits a wide range of very important pharmacological properties: analgesic, anti-inflammatory, anticancer, anticonvulsant, antibacterial, antiparasitic, and nutraceutical activity [81,82].

Hence, the diverse collection of terpenoid structures and functions triggers increased interest in their commercial use, resulting in some with well-established medical applications being registered as drugs on the market [81].

### 3.2. Nefroprotective Potential of Compounds from Amazonian Plant Species

Ischemic injury is a complex process of severe vasoconstriction and hypoxia, mainly in the renal cortex, with impairment of cellular integrity. The pathogenesis of ischemia and reperfusion injury involves multiple cellular and extracellular mechanisms [83].

Certain classes of plant compounds present in the Amazon, such as alkaloids, flavonoids, tannins, steroids, and terpenoids, have been investigated for their promising nephroprotective activity, including the modulation of different cellular mechanisms, thus promoting antioxidant activity, as well as anti-inflammatory activity [3].

The renal inflammatory process is characterized by an increase in chemotactic factors, including the chemokine protein chemotaxic protein-1 for monocytes (MCP-1) [84] and granulocyte and macrophage colony-stimulating factors (GM-CSF) [85]. Endothelial damage favors the formation of intercellular adhesion molecules (ICAM-1), adhesion molecules (VCAM), and P and E selectins, which promote leukocyte-endothelium interaction, platelet adhesion, and mechanical obstruction of the renal microvasculature. Monocytes cross the vascular endothelium and migrate to the damaged tissue, generating macrophages that produce inflammatory mediators, including transforming growth factor beta (TGF-β), tumor necrosis factor alpha (TNF-α), and interleukins 1, 6, and 12, Figure 1 [86].

CXC motif chemokine ligand 1 (CXCL1) is a cytokine belonging to the CXC subfamily of chemokines whose main receptor is CXC motif chemokine 2 (CXCR2), causing the migration and infiltration of neutrophils to sites of high expression [87]. CXCL1 plays a role in many adverse conditions associated with inflammation and neutrophil accumulation. Neutrophils phagocytose using reactive oxygen species (ROS), reactive nitrogen species (RNS), and other reactive small molecule compounds in affected tissues [88].

Factors such as pro-inflammatory cytokines, including interleukin-1β (IL-1β) and tumor necrosis factor α (TNF-α), act to increase the expression of CXCL1; these cytokines activate the nuclear factor κB (NF-κB), which also attenuated the increase of its expression [89]. The most significant CXCL1 receptor is CXCR2, a G protein-coupled receptor that acts in signal transduction by activating several signaling pathways such as extracellular signal-regulated kinase (ERK), mitogen-activated protein kinase (MAPK), and focal adhesion kinase (FAK) [90].

Toll-like receptors (TLR)—especially subtype 4 (TLR4) and the protein cluster of differentiation 14 (CD14) existing on the surface of monocytes, macrophages, dendritic cells, and neutrophils—act in the immunomodulatory activity [91]. TLR4 is responsible for initiating the production of inflammatory cytokines, increasing renal oxidative stress and macrophage-mediated inflammation, as well as activating the nuclear factor κB (NF-κB), which is essential in initiating the intrarenal inflammatory response in the event of nephrotoxicity [92,93].

NLR family pyrin domain containing 3 (NLRP3) are investigated for their association with chronic metabolic and inflammatory diseases [94]. This receptor regulates post-transcriptional processes, with formation of the inflammasome, composed of active caspase-1 and ASC adapter protein (an apoptosis-associated speck-like protein containing a CARD domain), which promote the cleavage of inactive pro-IL-1β interleukins and pro-IL-18 in their active forms [95,96].

NF-κB activation requires the transcription of specific genes that intervene in the encoding of inflammatory mediators, promoting immune, proliferative, anti-apoptotic, and anti-inflammatory responses [97]. This causes increased expression of tumor necrosis factor-α (TNF-α) in renal tubular cells, an important cytokine involved in systemic inflammation that coordinates the activation of a large network of pro-inflammatory cytokines, such as interleukin-1, 4, 6 (IL-1β, IL-4, IL-6), transforming growth factor β-1 (TGF-β 1), and the chemokine chemotaxic protein-1 for monocytes (MCP-1) [98,99].

The increased expression of renal injury molecule 1 (KIM-1), a tubular transmembrane protein, has a signaling function since it is associated with the activation of T cells and the immune response. When chronically expressed, it results in progressive renal fibrosis and chronic renal failure [100].

The inflammatory response with release of TNF-α, interferon-γ (IFN-γ), and IL-1 induces the expression of the enzyme inducible nitric oxide synthase (iNOS) in the renal medulla, in the glomerular mesangial cells, and in the endothelium cells of the renal vasculature [101]. With the release of inflammatory enzymes such as COX-2, whose transcription is dependent on NK-κB [102], as well as nitric oxide (NO), free radicals have the ability to interact with ROS and form toxic molecules such as peroxidonitrite, which oxidize and damage cell membrane proteins with even greater toxicity [101]. Therefore, phytochemical constituents found in Amazonian species may act to inhibit or attenuate these pathways.

The mitogen-activated protein kinases (MAPKs) signaling system consists of protein pathways with serine/threonine kinase activity. MAPKs are induced by cellular stress, inflammatory responses, and apoptotic pathways initiated by a variety of biological stressors [103]. MAPKs lead to the degradation of Iκβ (NF-κβ inhibitor); consequently, they act in the activation and migration of NF-κβ to the nucleus, producing pro-inflammatory cytokines including TNF-α [104].

Flavonoids and other classes of compounds also regulate many pathways such as MAPK, extracellular signal-regulated kinase (ERK), phosphoinositide 3 kinase (PI3K)/Akt, and protein kinase-related pathways to reduce oxidative stress and inflammation, potentially having nephroprotector effects [105].

Additionally, these natural compounds can reduce inflammation by acting on many regulatory substances. These include inhibition of NF-κB, activator protein-1 (AP-1), interleukin-1beta (IL-1β), tumor necrosis factor alpha (TNF-α), IL-6, IL-8, and COX2 [105,106].

Likewise, they act on promising and potent antioxidant molecules that confer anti-inflammatory activity, inducing transcription factors such as Nrf2, an antioxidant responsive element (ARE), which mediates the expression of antioxidant proteins. Nrf2 acts by suppressing the expression of MCP-1 and VCAM-1 and, thus, decreasing monocyte adhesion and transmigration to endothelial cells, which consequently reduces MAPK expression [107,108].

Therefore, mechanisms involving the activity of compounds such as alkaloids, flavonoids, tannins, steroids, and terpenoids may promote the ability to increase antioxidant enzymes, such as superoxide dismutase (SOD), chloramphenicol acetyltransferase (CAT), and plasma glutathione peroxidase (GSH-Px), as well as reduce the expression of inducible NO synthase (iNOS) and nitrites in the cell, thus protecting the renal cells [3,109].

### 3.3. Nefroprotective Potential of Amazonian Plant Species

There is a growing interest in the nephroprotective potential of Amazonian plant species. Several studies have investigated the potential of these species to prevent or treat kidney disease [110], Figure 2.

#### 3.3.1. *Banisteriopsis caapi* (Spruce ex Griseb.) Morton

*Banisteriopsis caapi* (Spruce ex Griseb.) Morton is a woody liana, common around the Amazon Basin, belonging to the kingdom Plantae, class Equisetopsida C. Agardh, order Malpighiales Juss. Former Bercht. & J. Presl, family Malpighiaceae Juss., genus *Banisteriopsis* CB Rob. and species *Banisteriopsis caapi* (Spruce ex Griseb.) Morton, which has broad ethnopharmacological use by the Amazonian people [111]. It is a species used as the main ingredient of the hallucinogenic drink, called ayahuasca, consumed by religious groups in Brazil to treat various ailments [112]. It is consumed for its hallucinogenic properties, which have been known by many of the indigenous people of the Amazon for centuries [113].

Some species of the Malpighiaceae family are known to produce alkaloids, among them, *B. caapi* (Spruce ex Griseb.) Morton. Harmine, one of the main alkaloids found in *B. caapi* (Spruce ex Griseb.) Morton, a plant widely consumed in the ayahuasca drink It is a β-carboline alkaloid widely disseminated due to its monoaminoxidase (MAO) inhibitory activity. As seen in the studies by Samoylenko et al. [114], harmine and harmaline (obtained from aqueous extracts of fresh and dried branches of *B. caapi* (Spruce ex Griseb.) Morton), have potent effects on MAO inhibitory and antioxidant activity. In addition, strong antioxidant activity for inhibition of cellular reactive oxygen species (ROS) generation by phorbol-12-myristate-13-acetate (PMA) has also been observed.

The effects of harmine against nicotine-induced damage in mouse kidneys were detailed by Salahshoor et al. [26]. In the study, administration of harmine to nicotine-treated animals significantly improved renal malondialdehyde (MDA), blood urea nitrogen (BUN), creatinine, and nitrite oxide levels. It increased the number of glomeruli and the level of power tissue ferric reducer/antioxidant (FRAP) compared to the nicotine group (*p* < 0.05), Table 1.

#### 3.3.2. *Peganum harmala* L.

The studied group belongs to the kingdom Plantae, class Equisetopsida C. Agardh, order Sapindales Juss. ex Bercht. & J. Presl, family Nitrariaceae Lindl., genus *Peganum* L. and species *Peganum harmala* L. [139]. The herbaceous *P. harmala* L. is perennial and branched, with leaves sectioned into three to five linear lobes. It produces whitish-yellow flowers and fruits in globular capsules with three chambers, containing black angular seeds [140]. It is commonly called wild rue, Syrian rue, or African rue [141].

Most species of the Nitrariaceae family contain alkaloids, which have been the subject of studies for their possible biological and pharmacological activity. For example, the studies by Niu et al. [142], when investigating the protective effect of harmine—the major compound isolated from *P. harmala* L.—in renal inflammation induced by lipopolysaccharide (LPS), as well as the respective molecular mechanisms involved, showed that pretreatment with harmine markedly alleviated the lesion kidney, reducing the release of renal biomarkers, inflammatory mediators, and the formation of malondialdehyde (MDA) and myeloperoxidase (MPO), while increasing superoxide dismutase (SOD) and glutathione (GSH) and reducing renal histopathological changes. Furthermore, in immunohistochemical staining and western blot analysis, the study indicated that the treatment with harmine suppressed the expression of the toll-like receptor 4 (TLR4), phosphorylation of nuclear factor kappa B (NF-κB) p65, and κBα inhibitor (IκBα), while the treatment also inhibited the expression of NLRP3, caspase-1, and interleukin-1β (IL-1β). In summary, pretreatment with harmine extracted from *P. harmala* L. can protect against LPS-induced acute kidney injury by attenuating oxidative stress and inflammatory responses and increasing antioxidant activity. The underlying mechanisms of harmine in mice with LPS-induced acute kidney injury may be related to the inhibition of the TLR4-NF-κB and NLRP3 pathways of the inflammasome.

Another study observed the effects of harmine on the renal activity of mice after cisplatin administration. The researchers demonstrated that there was a significant decrease in the total antioxidant capacity of the renal tissue, in the diameter of the renal corpuscles, and in the level of IL-10 expression in the group treated with cisplatin in relation to the control group, while the values of these parameters were significantly similar to those of the control group in the moderate or high dose groups treated with harmine + cisplatin. In addition, they noted significant increases in serum levels of urea and creatinine, Bowman’s space, amount of malondialdehyde, apoptosis rate, and gene expressions of TNF-α, NF-κB, IL-1β, and caspase-3 in the renal tissue of the cisplatin group compared to the control group, while these criteria did not differ in the harmine + cisplatin moderate or high dose groups. Thus, the study considered that harmine protected the kidneys against damage induced by cisplatin, and the antioxidant, anti-inflammatory, and anti-apoptotic properties of this compound were involved in the observed curative effect [118].

#### 3.3.3. *Passiflora edulis* Sims

The species under study belongs to the kingdom Plantae, class Equisetopsida C. Agardh, order Malpighiales Juss. ex Bercht. & J. Presl, family Passifloraceae Juss. ex Roussel, genus *Passiflora* L. and species *Passiflora edulis* Sims, with wide ethnopharmacological use by the people of the Amazon. *Passiflora edulis* Sims is a vine, supported by axillary tendrils [143]. It consists of palmate leaves, usually three-lobed with serrated margins; large flowers, with long peduncles, whitish, with a purple and pink triple crown; fruits, oval-shaped berries, containing abundant flat ovoid seeds, covered by a yellowish or brownish aril [144]. It has a vast geographic distribution: Brazil, Paraguay, Argentina, Antilles (West Indies islands), Central America, Venezuela, and Ecuador. It is commonly called passion fruit [121].

Pharmacological trials have shown numerous activity from compounds obtained from *P. edulis* Sims, including anxiolytic, sedative, neuropathic pain [118], activity linked to alcoholism and narcotics use [145], anticonvulsant and anxiolytic activity [119], cognitive function and degenerative diseases [120], antioxidant, antitumor action, hypoglycemic action, obesity, and insomnia [121].

The species is rich in natural bioactive compounds, among them, a significant content of flavonoids. For example, orientin and isoorientin are compounds with potential hypoglycemic effects, pointed out in the study by Galdino et al. [117] when evaluating the therapeutic effect of the aqueous extract of the fruit peel of *P. edulis* Sims as an adjuvant to insulin, to confer nephroprotection against diabetes induced by streptozotocin in Wistar rats. In the study, those animals treated with *P. edulis* extract showed superior glycemic control, which resulted in a reduction in the urinary albumin/creatinine ratio; maintenance of basal levels of mRNA expression of Nphs1, Nphs2, and Wt1n in the renal tissue; expression of mRNA Lrp2; prevention of protein loss from the renal tissue to the urinary space; and maintenance of glomerular basement membrane thickness, hyalinization, and glomerular and tubulointerstitial fibrosis with values close to those of the control group and significantly lower than those in the diabetic group. Therefore, the extract of *P. edulis* revealed potential therapeutic action of nephroprotection due to the reduction and prevention of the development of diabetic kidney disease.

The protective effect of flavonoids from *P. edulis* Sims was evaluated in alloxan-induced diabetic *Rattus norvegicus*, in which researchers observed renal dysfunction in uncontrolled diabetic groups, given the increased production of free radicals, with probable cellular damage and tubular damage, resulting in renal inflammation. In the study, the biomarkers urea and creatinine were measured in the animals’ bloodstream. Diabetic animals that received the flavonoid fraction of *P. edulis* Sims had lower urea and creatinine values when compared to the control group [146].

#### 3.3.4. *Annona muricata* L.

This plant species belongs to the kingdom Plantae, class Equisetopsida C. Agardh, order Magnoliales Bromhead, family Annonaceae Juss., genus *Annona* L. and species *Annona muricata* L. It has various uses in traditional indigenous medicine [139]. It is medium to large in size, reaching up to 10 m in height. Its leaves are green and shiny, with an oval shape and smooth texture, while its flowers are large and solitary, with thick, yellowish petals [147]. The species is widely distributed geographically, being found in several tropical regions of the world, including Central and South America, Africa, and Asia. It is known by several popular names, such as soursop, fruit of the count, and heart of queen, and is cultivated for its edible fruits, which have a sweet and slightly acidic taste [148].

In addition to its gastronomic uses, *A. muricata* L. is also consumed due to its medicinal properties. Many of these properties come from bioactive compounds present in its leaves, seeds, and fruits, with antioxidant, anti-inflammatory, antiparasitic, and anticancer activity [124].

A well-described example in the scientific literature is the species *A. muricata* L. This species has been studied for its bioactive metabolites, including acetogenins, and its constituents may have anticancer, hepatoprotective, neurotoxic, antinociceptive, antiulcerative, and chemopreventive activity [122].

A study of veterinary pharmacology and toxicology [149] demonstrated that *A. muricata* L. attenuates glycerol-induced nephrotoxicity in male albino rats through angiotensin-converting enzyme (ACE) signaling pathways. The methanolic extract of the leaves of *A. muricata* L., in that study, caused a significant decrease in the expression of the kidney injury molecule 1 (KIM-1) and exhibited antioxidant properties. This nephroprotective effect of the extract was observed by improving the levels of enzymatic and non-enzymatic antioxidants, suppressing inflammatory processes and inhibiting lipid peroxidation, thus revealing such antioxidant and anti-inflammatory properties.

#### 3.3.5. *Uncaria tomentosa* (Willd.) DC.

The group studied has wide ethnopharmacological use in the Amazon. It belongs to the kingdom Plantae, class Equisetopsida C. Agardh, order Gentianales Juss. ex Bercht. & J. Presl, family Rubiaceae Juss., genus *Uncaria* Schreb. and species *Uncaria tomentosa* (Willd.) DC. [150]. Preliminary phytochemical screenings demonstrated the marked presence of tannins in species belonging to the Rubiaceae family. These plant species are widely disseminated in the culture of traditional people and communities, due to their richness in the production of bioactive compounds [151,152].

Other phytochemical studies have found tetracyclic and pentacyclic oxindole alkaloids, indole and β-carbonyl alkaloids, flavonoids [153], coumarins [154], proanthocyanidins, steroids, ursan-derived triterpenoids, and quinovic acid glycosides [125].

The species *U. tomentosa* (Willd.) DC. has been associated with several health benefits, such as antioxidant and immunomodulatory, anti-inflammatory, analgesic and anticancer action, in addition to other medicinal properties. In traditional medicine, the plant is used to treat a variety of conditions, including infections, arthritis, diabetes, gastrointestinal problems, and “kidney cleansing” [126].

The renal benefits of herbal medicines such as *U. tomentosa* (Willd.) DC. have been demonstrated in the studies by Vattimo and Silva [155], when performing a pretreatment with *U. tomentosa* (Willd.) DC. in experimental models of ischemia/reperfusion, in which there was functional protection assessed by increased creatinine clearance, reduced peroxidation, and urinary thiobarbituric acid reactive substances (TBARS), probably related to the antioxidant activity of the herbal medicine.

#### 3.3.6. *Hymenaea courbaril* L.

The studied group belongs to the kingdom Plantae, class Equisetopsida C. Agardh, order Fabales Bromhead, family Fabaceae Lindl., genus *Hymenaea* L. and species *Hymenaea courbaril* L. [156]. The Fabaceae family is known to produce a wide variety of bioactive compounds, including tannins. The tannins present in *Hymenaea courbaril* L. have been the object of research for their potential biological effects, such as antioxidant, antiulcerogenic, anti-inflammatory, and antitumor properties [128]. Some studies also report the antiviral and antibacterial activity of these compounds [157].

*Hymenaea courbaril* L. has a wide distribution in South America and Central America; it is a large tree, reaching 15 to 20 m in height, and the trunk can be up to 1 m in diameter [158]. The flowers are pollinated by bats. Ripe fruits are eaten by rodents, birds, and monkeys, which, when breaking the fruits, release the seeds [159]. Its wood is considered valuable due to its high density and resistance to attack by xylophagous organisms. In the Amazon region, the species is known as jassaí, jataí, jataíba, jataíba stone, jataúba, jatel, jati, jatobá de anta, jutaí, jutaí açu, jutaí white, jutaí grande, and jutaí catinga [160].

The main phytochemical constituents found In the species are flavonoids, such as quercetin, kaempferol, and isorhamnetin, which are present in the leaves and fruits; tannins, such as catechins and proanthocyanidins, most commonly found in bark and seeds; fatty acids, including oleic and linoleic acid (seeds); and stilbenes, such as trans-resveratrol, found in the bark and fruit [161,162].

The tea produced from the bark of *H. courbaril* L. is indicated to treat kidney problems [163]. Pereira et al. [164] demonstrated the oxidizing activity of the methanolic fraction of *H. courbaril* L. seeds in mice treated with acetaminophen; the study showed the probable restoration of renal glutathione (GSH) levels in animals treated with the extract, in addition to reversing the increase in carbonylated proteins. Another study, using aqueous extracts of seed or bark of *H. courbaril* L., observed a reduction in renal levels of reactive substances to thiobarbituric acid 7 days after treatment [165].

#### 3.3.7. *Echinodorus macrophyllus* (Kunth) Micheli

This species belongs to the kingdom Plantae, class Equisetopsida C. Agardh, order Alismatales R. Br. ex Bercht. & J. Presl, family Alismataceae Vent., genus *Echinodorus* Rich. and the species *Echinodorus macrophyllus* (Kunth) Micheli, widely used by traditional medicine in Brazil [166]. The Alismataceae family is known for the presence of several bioactive compounds. Many Alismataceae species have traditionally been used in folk medicine for the treatment of a vast range of diseases, due to their diuretic and anti-inflammatory effects, as well as in kidney and liver disorders [131]. Some of the bioactive compounds present in plants of this family, such as *Echinodorus macrophyllus*, have been the object of scientific investigation for their pharmacological properties and possible therapeutic applications [167].

*Echinodorus macrophyllus* is a perennial plant, herbaceous or subshrub, of aquatic origin, emerging from the water. It has rhizomes and can reach between 1 and 2 m in height. Its leaves are petiolate and oval, with a heart-shaped base and a sharp tip [168]. Popularly, the species is known as leather hat, water hyacinth, campanha tea, brejo tea, poor man’s tea, mineiro tea, congonha do brejo, brejo herb, and swamp herb [169].

According to Silva et al. [129], among the constituents produced by the species are the terpenic profile containing linalool, α- and β-caryophyllene, *E*-nerolidol, and phytol as predominant, as well as a variety of diterpenoids belonging to the same classes, such as the chapecoderines of the group labdanos. Furthermore, a (+)-3-carene derivative was detected, along with a significant proportion of carotenoids. Gasparotto et al. [130] demonstrated the presence of the flavonoids vitexin and isovitexin. While Garcia et al. [170] found the presence of phenylpropanoids in the species, such as ferulic and *E*-caffeoyl tartronic acid (2-*E*-caffeoyloxymalonic acid).

Traditionally, the Amazonian population uses extracts from the leaves of *E. mac-rophyllus*, from infusion, decoction, or maceration methods, in water or alcohol, to treat urinary system disorders, as they are known to be powerful diuretic agents. In view of popular usage, Nascimento et al. [171] demonstrated that preconditioning with *E. macrophyllus* attenuated cyclophosphamide-induced acute kidney injury in rats as evidenced by increased creatinine clearance and reduced oxidative metabolites in urine and increased reserve of antioxidant enzymes in renal tissue.

Studies carried out in a model of acute kidney injury induced by gentamicin found similar results of antioxidant protection of *E. macrophyllus*, when administering crude ethanolic extracts of leaves and fractions of *E. macrophyllus* by endogastric route, in normal rats or with acute tubular necrosis induced by gentamicin-cine. Thus, it was demonstrated that it produced a dose-dependent reduction in urine output. The extracts in question were effective in reversing all changes induced by gentamicin, such as polyuria and reduction in the glomerular filtration rate; in addition, the morphological changes induced by gentamicin were not observed in animals that were treated with extracts of *E. macrophyllus* concomitantly with gentamycin [167].

#### 3.3.8. *Acmella oleracea* (L.) R. K. Jansen

The studied group belongs to the kingdom Plantae, class Equisetopsida C. Agardh, order Asterales Link, family Asteraceae Bercht. & J. Presl, genus *Acmella* Rich. ex Pers. and the species *Acmella oleracea* (L.) R. K. Jansen, which is widely used in medicine and cooking by traditional Amazonian people [172]. The Asteraceae family is known for the presence of bioactive compounds, such as sesquiterpene lactones and flavonoids [173]. Primarily for their medicinal properties, species in this family have traditionally been used to treat a wide range of ailments, including respiratory problems, inflammation, headaches, gastrointestinal problems, and infectious diseases [24]. Scientific research has focused on some of the bioactive compounds present in plants of this family, such as *Acmella oleracea*, in search of possible therapeutic applications, such as analgesic, anti-inflammatory, antimicrobial, and antioxidant activity [136].

*Acmella oleracea* is an important medicinal herb, which occurs in tropical and subtropical regions of the planet. It is an annual, perennial herbaceous, 30–40 cm high, semi-straight, or creeping, with cylindrical, a fleshy stem and decumbent branches, usually without roots at the nodes. The main root is pivotal, with abundant lateral branches. The leaves are opposite, membranous, and petiolate [174]. The species is popularly known as jambu, cress from pará, abecedária, cress bravo, cress from Brazil, cress from the north, buttercup, crazy herb, jabuaçu, and nhambu [136].

*Acmella oleracea* is used in northern Brazil for the treatment of various diseases, such as tuberculosis, flu, cough, and rheumatism, and as an anti-inflammatory; in addition, hydroethanolic formulations with this species are popularly used as a female aphrodisiac, for treatment of male sexual dysfunctions, and as a diuretic [135,136].

Regarding the production of metabolites, *A. oleracea* is a rich source of secondary metabolites, and its phytochemistry has been widely investigated [175]. Borges et al. [176] observed an increase of 31.6% in the content of spilanthol and 16.8% of flavonoids in the inflorescences and higher contents of total phenols, carotenoids, spermidine, and spermine in the leaves and flowers of jambú. The work by Abeysiri et al. [132] revealed that alkaloids, flavonoids, saponins, steroid glycosides, and tannins are distributed in all parts of the plant. Going into more detail about the phytochemical composition of *A. oleracea*, several triterpenoids were found, such as 3-acetylaleuritolic acid, β-sitostenone, and stigmasterol. Furthermore, steroidal glycosides, namely, stigmasteryl-3-O-β-D-glucopyranoside and β-sitosteryl-3-O-β-D-glucopyranoside, have been identified. Several phenolic compounds were also detected, such as vanillic, trans-ferulic, and trans-isoferulic acids; scopoletin; and fatty acids such as n-hexadecanoic and n-tetradecanoic acids [132,133,134].

Some studies have observed a marked diuretic action of aqueous extract of *A. oleracea* inflorescences in rats; the authors have described an increase in Na^+^ and K^+^ levels and a reduction in osmolarity in the urine of animals treated with the extract [177]. Yadav and collaborators [178] showed that the ethanolic extract of *A. oleracea* in rats provided diuresis similar to that produced by the action of furosemide. Gerbino et al. [179] consider that the inhibition of cyclic AMP induced by spilanthol negatively modulates the mechanisms of urine concentration. Furthermore, the mechanisms of action on the kidney show that *A. oleracea* is a promising source of compounds with diuretic activity.

#### 3.3.9. *Rosmarinus officinalis* L.

The species belongs to the kingdom Plantae, class Equisetopsida C. Agardh, order Lamiales Bromhead, family Lamiaceae Martinov, genus *Rosmarinus* L. and species *Rosma-rinus officinalis* L.; it has wide ethnopharmacological use [180]. The Lamiaceae family is one of the most important herbaceous families; it is composed of an immense variety of plant species with biological and medicinal applications [138]. This family includes numerous aromatic spices, including *Rosmarinus officinalis* L., a plant species commonly known as rosemary, which is useful in cooking due to its characteristic aroma; it is widely used by indigenous populations where it grows spontaneously [181].

*Rosmarinus officinalis* is a shrubby herb, widely used in culinary, medicinal, and commercial applications, including the fragrance and food industries [182]. The leaves (fresh or dried) are consumed due to the characteristic odor that they offer to the dish. They are also consumed in small amounts in the form of tea, while extracts of *R. officinalis* are regularly used for their active natural antioxidant properties to improve shelf life of perishable foods [183].

Phytochemical screenings carried out on the species revealed 0.5% to 2.5% volatile oil in the leaves. Among the phytocompounds, the species exhibits the presence of monoterpene hydrocarbons (alpha and beta-pinene), camphene, limonene, camphor (10% to 20%), borneol, cineol, linalool, and verbinol [137]. In addition to numerous volatile and aromatic components, the species has flavonoids, such as diosmetin, diosmin, genkwanin, luteolin, hispidulin, and apigenin, as well as terpenoid compounds such as triterpenes (oleanolic and ursolic acid) and diterpene carnosol. Among the phenols found in the species are caffeic, chlorogenic, labiatic, neochlorogenic, and rosmarinic acids, as well as a considerable number of salicylates [182,183,184].

Among the ethnomedicinal applications for *R. officinalis* are analgesic, anti-inflammatory, anticarcinogenic, antirheumatic, nephroprotective, spasmolytic, antihepatotoxic, atherosclerotic, carminative, and choleretic action. It also offers protection against UV and gamma radiation and improvement of stress [138].

Zohrabi et al. [185] investigated the effect of an oral extract of *R. officinalis* on acute renal failure (ARF) disorders induced by ischemia/reperfusion in rats. The authors showed that the aqueous extract of *R. officinalis* suffered the oxidative stress marker malondialdehyde (MDA), increased the ferric antioxidant reducing power (FRAP) compared to the vehicle groups and, regarding the histopathological analyses, observed a significant reduction in vessel management, disturbance of the tubules, and Bowman’s Capsule space compared to the vehicle groups.

Another study evaluated the effectiveness of *R. officinalis* essential oil (REO) against changes induced by potassium dichromate in the kidneys of male rats, in which they injected hexavalent chromium to induce renal dysfunction (oxidative damage and alterations in the antioxidant defense system, and histopathological and immunohistochemical alterations). The animals were treated with REO before or after the induction of renal dysfunction, resulting in an improvement in the toxic effect by extinguishing, chelating, and detoxifying free radicals and enhancing the state of antioxidant defense [186].

## 4. Conclusions

Knowing the phytocompounds with potential nephroprotective effects against AKI based on the traditional Amazonian knowledge of treating different ailments that disturb/affect the health of the kidneys is generally passed on over generations by healers, housewives, and elderly people from riverside communities, who, due to limited access to health services, use this precious information about the natural resources of the Amazon as their only resource. The pharmacotoxicological validation of this information is highly necessary, considering that it subsidizes the knowledge of the medicinal potential of the Amazonian flora, substantially improving the phytochemical and pharmacological relevance of these species, especially in the face of AKI, a clinical condition with high morbidity and mortality. Although much of the research on the nephroprotective potential of Amazonian plant species is still in the preclinical stage, these plants show promise as a potential source of new therapies for kidney disease. However, more research is needed to fully understand its mechanisms of action and possible side effects, as well as to develop safe and effective dosages for human use.

## Figures and Tables

**Figure 1 molecules-28-06411-f001:**
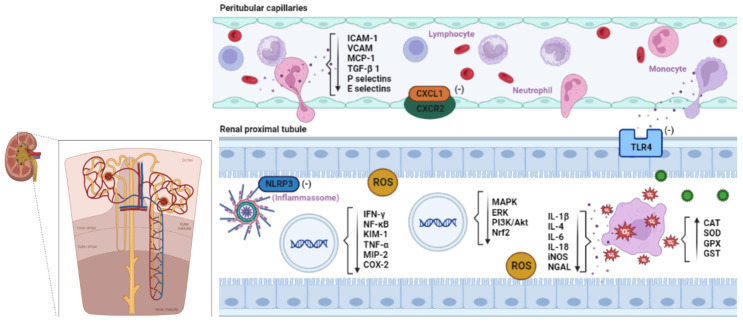
Amazonian phytocompounds against acute kidney injury. The proposed mechanism applies mainly to the kidneys. Classes of Amazonian plant compounds such as alkaloids, flavonoids, tannins, steroids, and terpenoids have significant biological activity especially for the treatment of acute kidney injury (AKI). Endothelial injury in induced AKI favors the formation of intercellular adhesion molecules (ICAM-1), adhesion molecules (VCAM), chemokine chemotaxic protein-1 for monocytes (MCP-1), transforming growth factor β-1 (TGF-β 1), and P and E selectins, which promote leukocyte–endothelium interaction, platelet adhesion, and mechanical obstruction of the renal microvasculature; however, active biological substances can act in the expressive reduction of these chemokines. CXC motif chemokine ligand 1 (CXCL1) is a cytokine belonging to the CXC subfamily of chemokines, whose main receptor is CXC motif chemokine 2 (CXCR2), a G protein-coupled receptor that causes neutrophil migration and infiltration. Activation of CXCR2 results in signal transduction through multiple pathways. Flavonoids and tannins act to reduce this expression; pro-inflammatory cytokines, such as interleukin-1β (IL-1β) and tumor necrosis factor α (TNF-α), activate the nuclear factor κB (NF-κB), which increases the expression of the CXCL1 gene. CXCL1/CXCR2 mediates the activation of phosphatidylinositol-4,5-bisphosphate 3-kinase (PI3K), and C-β phospholipase (PLC-β). PI3K induces activation of protein kinase B (PKB)/Akt, and signals pathways including extracellular signal-regulated kinase (ERK), mitogen-activated protein kinase (MAPK), and focal adhesion kinase (FAK). Toll-like receptor 4 (TLR4) is responsible for initiating the production of inflammatory cytokines; its inhibition results in decreased inflammation and renal dysfunction during nephrotoxicity. The NLRP3 receptor regulates post-transcriptional processes, which lead to the formation of inflammasome, responsible for the maturation of the inactive forms pro-IL-1β and pro-IL-18; the decrease of its expression leads to the inactivation of macrophages and lymphocytes. The activation of NF-κB promotes the transcription of specific genes that encode inflammatory mediators; however, the exogenous induction of active compounds decreases the expression of inflammatory mediators such as interferon-γ (IFN-γ), which leads to a reduction in the expression of renal injury molecule 1 (KIM-1), tumor necrosis factor-α (TNF-α) in renal tubular cells, and consequently the inactivation of a large network of pro-inflammatory cytokines, such as interleukin-1, 4, 6 (IL-1β, IL-4, IL-6, IL-18), macrophage inflammatory protein 2 (MIP-2) chemokines, among others. It also induces a decrease in the expression of the inflammatory enzyme cyclooxygenase (COX-2) and oxide nitric synthase (iNOS), in the renal medulla, in the glomerular mesangial cells, and in the endothelial cells of the renal vasculature, thereby reducing the tubular stress marker lipocalin associated with neutrophil gelatinase (NGAL). The signaling system of mitogen-activated protein kinases (MAPKs) is induced by cellular stress, by inflammatory responses. MAPK activation leads to the degradation of Iκβ (NF-κβ inhibitor), consequently promoting NF-κβ activation and migration to the nucleus. Flavonoids, tannins, steroids, and terpenoids also regulate many pathways such as MAPK, extracellular signal-regulated kinase (ERK), phosphoinositide 3 kinase (PI3K)/Akt, and related protein kinase pathways to reduce oxidative stress and inflammation, and they may have nephroprotective effects. These compounds also induce transcription factors such as Nrf2, an antioxidant responsive element (ARE), which mediates the expression of antioxidant proteins. Nrf2 suppresses MCP-1 and VCAM-1 expression and thus decreases monocyte adhesion and transmigration to endothelial cells, which reduces MAPK expression. These mechanisms enhance the activity of endogenous antioxidants such as catalase (CAT), superoxide dismutase (SOD), glutathione peroxidase (GPX), and glutathione S-transferase (GST), which act together to provide a line of defense against oxidative damage.

**Figure 2 molecules-28-06411-f002:**
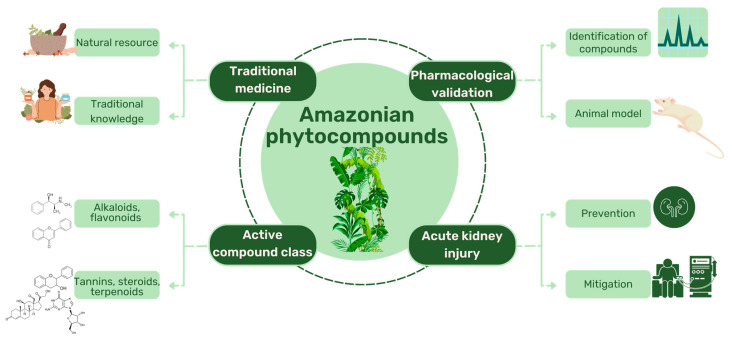
Potential of Amazonian phytocompounds against acute kidney injury.

**Table 1 molecules-28-06411-t001:** Phytocompounds from Amazonian plant species and their pharmacological activity.

Species	Parts Used	Isolated or Characterized Constituents	Pharmacological Activity
*Banisteriopsis caapi* (Spruce ex Griseb.) Morton	Stem	Harmine **(1)**, harmaline **(2)** [114], tetrahydroharmine **(3)**, and harmalinic acid **(4)** [115]	Analgesic [22], hallucinogen [23], anesthetic [24], antidiabetic [25], anticancerogenic [18], nephroprotective, diuretic [26]
*Peganum harmala* L.	Seeds	Harmol **(5)**, harmalol **(6)**, harmine **(1)**, and harmaline **(2)** [116]	Antioxidant, nephroprotective, anti-inflammatory, anti-apoptotic [116]
*Passiflora edulis* Sims	Fruit peel, leaves, flowers, seeds	Orientin **(7)** and isoorientin **(8)** [117]	Anxiolytic, sedative, neuropathic pain [118], anticonvulsant [119], cognitive function and degenerative diseases [120], antioxidant action, antitumor action, hypoglycemic action, obesity, insomnia, nephroprotector [121]
*Annona muricata* L.	Leaves	Acetogenin **(9) [122]**, *δ*-Cadinene **(10)**, and *α*-Muurolene **(11)** [123]	Anticancerogenic, hepatoprotective, neurotoxic, antinociceptive, antiulcerative, chemopreventive, nephroprotective [124]
*Uncaria tomentosa* (Willd.) DC.	Stem	Uncarine F **(12)**, speciophylline **(13)**, and mitraphylline **(14)** [125]	Antioxidant and immunomodulator, anti-inflammatory, analgesic, anticancer, and diuretic [126]
*Hymenaea courbaril* L.	Stem and leaves	Fisetin **(15)**, cyclosativene **(16)**, caryophyllene **(17)**, and α-himachalene **(18)** [127]	Antioxidant, antiulcerogenic, anti-inflammatory, antitumor, and diuretic [128]
*Echinodorus macrophyllus* (Kunth) Micheli	Leaves	Linalool **(19)**, α-caryophyllene **(20)**, β-caryophyllene **(21)** [129], isovitexin **(22)**, and isoorientin **(8)** [130]	Diuretic, anti-inflammatory, treatment of kidney and liver disorders [131]
*Acmella oleracea* (L.) R. K. Jansen	Flowers and leaves	Spilanthol **(23)**, spermidine **(24)**, spermine **(25),** and 3-acetylaleuritolic acid **(26)** [132,133,134]	Aphrodisiac, treatment of male sexual dysfunctions, diuretic, and anti-inflammatory [135,136]
*Rosmarinus officinalis* L.	Leaves	Camphene **(27)**, limonene **(28)**, camphor **(29)**, borneol **(30)**, cineol **(31),** and linalool **(19)** [137]	Analgesic, anti-inflammatory, anticarcinogenic, antirheumatic, nephroprotective, spasmolytic, antihepatotoxic, atherosclerotic [138]
**Harmine** 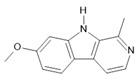 **(1)**	**Harmaline** 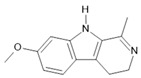 **(2)**	**Tetrahydroharmine** 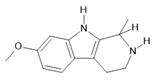 **(3)**	**Harmalinic acid** 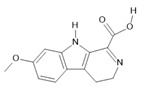 **(4)**	**Harmol** 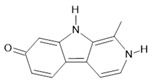 **(5)**
**Harmalol** 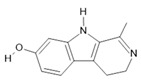 **(6)**	**Orientin** 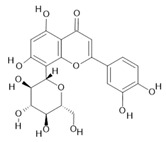 **(7)**	**Isoorientin** 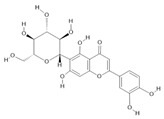 **(8)**	**Acetogenin** 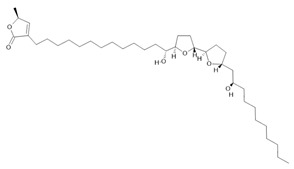 **(9)**
***δ*-Cadinene** 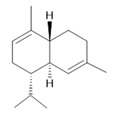 **(10)**	***α*-Muurolene** 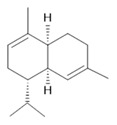 **(11)**	**Uncarine F** 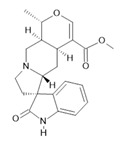 **(12)**	**Speciophylline** 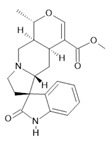 **(13)**	**Mitraphylline** 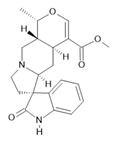 **(14)**
**Fisetin** 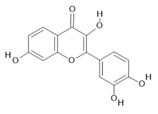 **(15)**	**Cyclosativene** 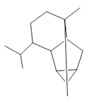 **(16)**	**Caryophyllene** 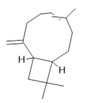 **(17)**	**α-himachalene** 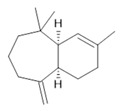 **(18)**	**Linalool** 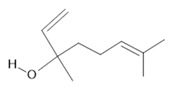 **(19)**
**α-caryophyllene** 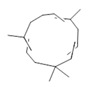 **(20)**	**β-caryophyllene** 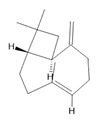 **(21)**	**Isovitexin** 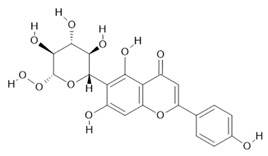 **(22)**	**Spilanthol** 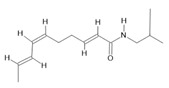 **(23)**
**Spermidine** 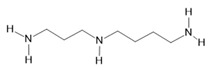 **(24)**	**Spermine** 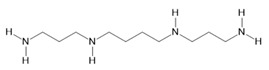 **(25)**	**3-acetylaleuritolic acid** 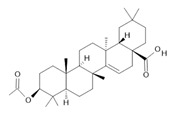 **(26)**	**Camphene** 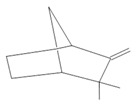 **(27)**
**Limonene** 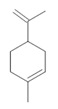 **(28)**	**Camphor** 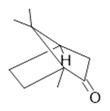 **(29)**	**Borneol** 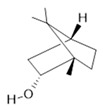 **(30)**	**Cineol** 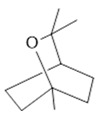 **(31)**	

The numbers in bold correspond to the molecular structures shown below.

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
