# Peer review of "Phytocompounds from Amazonian Plant Species against Acute Kidney Injury: Potential Nephroprotective Effects"

_molecules, 2023, doi:10.3390/molecules28176411_

Round 1

Reviewer 1 Report

Dear Authors

The MS entitled “Phytocompounds from Amazonian Plant Species against Acute 2 Kidney Injury: Potential Nephroprotective Effects” was thoroughly evaluated. The MS discuss the potential use of some Amazonian medicinal plants, their main constituents and potential in remedy against kidney related problems. The study summarizes some of the pharmacological evidences of such plants against animal models and some preliminary investigations in humans. The following suggestions and amendments should be incorporated in the MS.

General corrections.

·         The MS should be revised for format corrections (line spacing etc.)

·         Line 108: Scholar Google should be Google scholar.

·         Line 506: kindly make corrections where superscript or subscripts are involved such as Na+ should be Na+.

Specific:

·         A sunburst diagram should be added that could summarize the main active constituents, their host plants and the particular kidney problem/ disease that is targeted. This will increase a comprehensiveness in this study.

·         Line 147: there are certain alkaloids which cause kidney problems and a lot of literature is also available. The toxicity of plants towards kidney problems should also be discussed as not all the natural products are healing agents. Particularly, the toxicity of certain metabolites (alkaloids/saponins) could produce acute renal toxicity in humans. 

·         Do we have any particular prescribed use of such plants products/compounds as medicine to cure kidney related diseases? If any, please mention in the MS.

·         Is there any particular correlation between antioxidant potential towards kidney treatments? Especially molecular basis. Add these in flavonoids importance.

Dear Authors

The MS entitled “Phytocompounds from Amazonian Plant Species against Acute 2 Kidney Injury: Potential Nephroprotective Effects” was thoroughly checked for english language. The punctations, grammer and scientific language is OK. Kindly revised your abstract and make it concise.  

Author Response

All comments were met and we are forwarding the manuscript with the highlighted corrections.

Reviewer 2 Report

Dear Authors,

It was a pleasure to get familiar with your manuscript. It is an interesting effort to combine traditional medicine with pharmacological validation.

Various classes of secondary methabolites, examples of plant species and a pathophisiology of AKI were comprehenvisely described. 

Currently, emphasis is placed on compounds of natural origin, so this review give some indications on medicinal plants/secondary methabolites that may be promising in the fight against AKI. For this and other reasons mentioned above I reccommend publication of your work in Molecules.

Best regards,

Reviewer

Author Response

(The authors gave the same response as above.)

Reviewer 3 Report

Dear Authors:

Here you have some comments for improving your paper. 

Abstract:

Line 21-23: You write "this review aimed to describe the pharmacological properties of phytocompounds from Amazonian plant species and their effectiveness in the prevention and treatment of Acute Kidney Injury. (AKI)."

Please check this aim to be more clear and reflect the review.

Keywords:

Line 29: Maybe you should include Amazonian traditional medicine or only Amazonian?. You are including the clinical condition and its pharmacological activities??

Introduction:

Line 104-125: In this section you also describe the methodology of the review?? Don't you think it is better to describe clearly and specifically in another section??

You should include the reserach question, search strategy (keywords used referring to phytocompounds or secondary metabolites, plant species, condition and search period); databases used; in/out criteria and data extraction. You should also mention the number of results, the number of excluded and included results and the reason for that.

How do you select the plant species??

What kind of study do you search for a nephroprotective effect??? in vitro, in vivo or clinical studies???

Secondary metabolites:

Line 128: from now on you describe the phytocompounds,  but you need to focus on AKI and its mechanisms of action.

Maybe you can include a table with all secondary metabolites found for AKI and its mechanisms of action, as the main objective is phytocompounds.

Amazonian plants:

Line 236: from now on you explain 9  amazonian species to treat AKI. How do you select these species???

In Figure 1 you show animal model as one of pharmacological validation, it means that your review is focused on animal/in vivo studies???

In Table 1 you include 9 species with its pharmacological activities. Does its isolated or characterized constituents have all these pharmacological activities??

It would be interesting to give info about the type of phytochemicals (alkaloids, flavonoids, etc.) with the chemical structures.

Nephroprotective effect:

Line 555: In the section Nefroprotective Potential of Compounds from Amazonian Plant Species, what info do you want to show??? Mechanisms of action of secondary metabolites??? Don't you think it is better to include these info in secondary metabolites section???

Conclusions:

Line 692: Please outline clearly the conclusion.  Do you have one or several conclusions? This would be in line with your objective/s.

Good luck!!

Author Response

(The authors gave the same response as above.)

Round 2

Reviewer 1 Report

The MS is Ok however, the  sunburst diagram have not been added.

Reviewer 3 Report

Dear Authors,

well done.